# An Opportunity to Fill a Gap for Newborn Screening of Neurodevelopmental Disorders

**DOI:** 10.3390/ijns10020033

**Published:** 2024-04-16

**Authors:** Wendy K. Chung, Stephen M. Kanne, Zhanzhi Hu

**Affiliations:** 1Department of Pediatrics, Boston Children’s Hospital, Boston, MA 02115, USA; 2Harvard Medical School, Boston, MA 02115, USA; 3Department of Psychiatry, Weill Cornell Medical College, New York, NY 10065, USA; smk4004@med.cornell.edu; 4Department of Systems Biology, Columbia University Irving Medical Center, New York, NY 10032, USA; zh2425@cumc.columbia.edu

**Keywords:** newborn screening (NBS), genome sequencing (GS), neurodevelopmental disorder (NDD), variants of unknown significance (VUS), early intervention program (EIP), genomic uniform screening against rare diseases in all newborns (GUARDIAN)

## Abstract

Screening newborns using genome sequencing is being explored due to its potential to expand the list of conditions that can be screened. Previously, we proposed the need for large-scale pilot studies to assess the feasibility of screening highly penetrant genetic neurodevelopmental disorders. Here, we discuss the initial experience from the GUARDIAN study and the systemic gaps in clinical services that were identified in the early stages of the pilot study.

## 1. Introduction

Newborn screening (NBS) has long been one of the most successful public health programs and has saved lives and decreased disease burdens worldwide. Consented prospective NBS research pilot studies have the potential to accelerate therapeutic development for the thousands of genetic diseases currently without treatments as they can identify presymptomatic patients before progressive damage occurs and when treatments can be most effective. The pilot newborn screening studies of Spinal Muscular Atrophy (SMA) that were performed in tandem with clinical trials for novel treatments for SMA provide evidence that strategically planned implementation of expansions to newborn screening can rapidly impact the clinical outcomes for rare genetic diseases at a population level [1].

In recent years, considering expanding NBS using DNA sequencing has been made possible with the decreasing cost of sequencing, the advancement in genetic variant interpretation methods and reference databases, and the emerging therapeutic modalities for genetic diseases, such as gene/cell therapies and antisense oligonucleotide treatments. Among the conditions that can be accurately identified by DNA sequencing are over 500 neurodevelopmental disorders (NDD) of high penetrance and early onset, which have a significant impact on patients and families [2]. As the number of monogenic NDD conditions continues to expand, exome/genome sequencing (ES/GS) has been suggested as a screening platform to enable the rapid and iterative evolution of the conditions screened as new conditions are described and new treatments become available. Early recognition of neurodevelopmental conditions can have benefits in terms of the initiation of early intervention and recognition of associated medical conditions, including seizures and congenital anomalies [3,4]. However, this group of conditions has generally not been considered for NBS for a number of reasons, such as a lack of evidence regarding effective treatments, complexities in accurately predicting prognosis, and a lack of consensus on the benefits of early diagnosis. For all these reasons, some parents may reasonably decline learning about potential neurogenetic diagnoses regarding their newborns.

## 2. Previous Explorations

In 2021, the Simons Foundation Autism Research Initiative (SFARI) organized a Newborn Screening Workshop to explore a possible framework for a pilot study to screen NDDs in newborns using GS [5]. Although many challenges were identified, it was also recognized that adding highly penetrant NDDs to NBS has potential value. It helps to avoid stressful and costly diagnostic journeys and enables equitable access to diagnoses. It provides the possibility of earlier interventions, including seizure control, identification of associated hearing and vision deficits that are correctable, and enrollment into Early Intervention Programs (EIP) [6] to support early motor and language development and potentially improve social cognition. It empowers families with information to manage medical care and access to a community of families and providers with information about their child’s condition and can catalyze the development of novel treatments for these rare conditions. A multi-stakeholder public–private partnership approach was recommended to ensure an equitable, balanced, and responsible piloting of population-based screening.

## 3. GUARDIAN Study and the Systemic Gaps Experienced

The GUARDIAN study (Genomic Uniform screening Against Rare Diseases In All Newborns) was launched in 2022 as a multi-site single-arm prospective investigation of supplemental newborn genomic screening. The study focused on genetic conditions in two groups. Group 1 was composed of 156 effectively treatable conditions, and all the consenting participants underwent screening. The optional group 2 was composed of 99 NDDs, most associated with seizures in some individuals with the condition, with expected benefits of early treatments and interventions. Positive screening results were followed by additional testing to conduct final diagnoses, including parental genetic testing to determine inheritance/de novo status and clinical assessments for associated features. The ongoing study has screened over 10,000 babies, with ~91% of the enrolled babies also consented to screen for group 2 (unpublished data). However, while the parents have been satisfied with their decision to enroll in GUARDIAN based upon surveys after return of results, one significant post-screening challenge to address has emerged.

The benefits of early identification and intervention of NDDs have been documented [3,7,8,9]. The expanded Individuals with Disabilities Education Act (IDEA Part C) aimed to establish a comprehensive system with federal funding and implementation at the state level for delivering early intervention services for children up to age 3 years [6]. States were required to develop their own criteria for eligibility, definitions of developmental delay, and financial priorities to meet or surpass the federal requirements. Consequently, there is significant variation among states, which was further complicated by states not having reciprocity. Thus, although publicly funded EIPs are in place nationwide, determining eligibility for new patients with different conditions and onboarding to care vary by local policy and evaluator. Conventionally, eligibility for EIP services is based on an evaluation of the child’s development and abilities. Patients referred to EIPs are usually clinically symptomatic and referred with developmental concerns. The evaluations assess age-specific development, and the services generally require a recognizable delay of 25–30%. However, it is nearly impossible to accurately assess delays in infants identified within the first few weeks of life, and the evaluators and assessment tools are not sensitive to the small differences in manifestations in infants. It is for this reason that common genetic conditions associated with NDD, such as Down syndrome, often afford newborns automatic eligibility. Because Down syndrome has been recognized for decades and is relatively common and often prenatally or neonatally diagnosed with noninvasive prenatal screening and physical examinations, respectively, there are well-accepted clinical care guidelines that recommend early intervention for Down syndrome from birth [10]. It is important to note that, as of 2019, only 32 states included Down syndrome in their list of conditions eligible for EIP services. This variation in inclusion criteria across states reflects the wide diversity in state policies regarding eligibility [11].

In contrast to Down syndrome, the majority of the conditions in GUARDIAN are not specifically included for automatic eligibility, in part because many of these conditions have only been identified within the last decade with the advent of clinical genomic sequencing methods. The impact of various NDDs on development, including when and to what extent the impact manifests, can vary significantly. This variability makes it challenging for early intervention agencies to justify providing support solely based on a diagnosis without an evaluation confirming a current delay significant enough to warrant intervention. If infants are not immediately eligible for services, families are forced to wait for re-evaluation. This situation can be frustrating for families who find themselves waiting for the impact of the disorder to reach a substantial enough level before receiving an intervention. The frustration is compounded by having to wait for an intervention despite evidence of emerging delays and literature supporting the benefits of early intervention, including parent-mediated interventions [12,13]. With the current system, we are missing the opportunity to support the window of early brain development that we intended to target.

A unique example of such NDD conditions is Rett syndrome, which affords specific opportunities and challenges. Rett syndrome is estimated to affect ~7 in 100,000 girls [14]. Early development is normal; however, development regresses starting at around 12 to 18 months of age [15]. A lack of distinguishing physical features has historically led to diagnoses only after evidence of developmental regression. Encouragingly, animal data suggest that early intervention with motor and memory training prior to the onset of symptoms improves outcomes [16], and clinical trials of genetic therapy for MECP2 are currently open for older individuals (Clinical trial NCT05898620). However, with the current policies for EIP, girls with *MECP2* genetic diagnoses might need to wait until 24 months of age to qualify for services. Another NBS pilot program (Early Check) has encountered similar hurdles, where presymptomatic infants were diagnosed with Fragile X syndrome (another rare NDD condition) but were unable to obtain early intervention support (A. Wheeler and S. Scott, personal communication).

Furthermore, even for infants who qualify for EIP, identifying effective service providers can be challenging because most are not experienced with infants. While the parents/guardians/caregivers are motivated to support their young child, they are frustrated with their inability to access support and services at the same time they are coping with a mixture of emotions, including sadness, anxiety, and sometimes denial, as they adjust to what can be a serious diagnosis.

To address this frustration and to support children, we are encouraging families to proactively participate in evidence-based enrichment programs designed to support the development of infants and toddlers. One such program is the “Help is in Your Hands” course, rooted in the evidence-based Early Start Denver Model (ESDM) [17] and Joint Attention, Symbolic Play, and Engagement Regulation (JASPER) [18] curriculum, which specifically targets those at risk for autism. Other more general resources include the online tool My Baby Navigator from Florida State University (https://my.babynavigator.com/, accessed on 27 February 2024) and the Caregiver Skill Training by the World Health Organization (https://openwho.org/courses/caregiver-skills-training, accessed on 27 February 2024), which is applicable to various NDDs. It is important to acknowledge that these programs may pose additional challenges for already overwhelmed parents. Additionally, some programs may not specifically cater to infants younger than 18 months. Despite these drawbacks, these programs serve as valuable educational tools for parents and can help to partially fill the gap in service provision.

## 4. Discussion

The promise of newborn screening relies on effective treatment before symptom onset, which clashes with the traditionally structured medical care systems in which treatments are set up to be administered only after symptoms are observed. Here, we have identified a systemic gap in the care system of EIP, which was established to only support children already symptomatic with NDDs. It was not considered that we would be able to identify infants at high risk prior to symptom onset. The current system leads to delays in providing supporting therapies and missed opportunities to improve outcomes.

The frontier of clinical neuroscience is advancing rapidly, and novel methods of genetically based therapy with new strategies for delivery to the brain suggest that, over the next decade, there will be therapeutic or even curative opportunities for many neurogenetic conditions. Diagnosing these conditions early enough for treatment to be effective will be especially important as we prepare for clinical trials of genetically based treatments. Thus, there will be a critical need for the field to have systems in place to allow families to access genomic screening for genetic NDDs for a period of time before the healthcare infrastructure catches up if we want to accelerate treatments for these conditions and support greater equity in access to care. As genetic diagnoses of NDDs are increasingly being conducted through clinical care prenatally and neonatally, the federal and state government should consider establishing a core list of NDDs that fit the same phenotypes as commonly covered genetic conditions such as Down syndrome, as well as providing resources to expand the automatic provision of EIP services to these conditions, with periodic reassessment of the list. For NDDs that are not included in the list due to insufficient evidence to support associations with significant neurodevelopmental disabilities, research funding agencies should consider using a portion of their grant funding to support infants while evidence regarding these conditions accumulates. Studies of these rare genetic conditions will require natural history research to document the developmental course to provide the evidence necessary to evaluate the appropriateness of inclusion for services. Consensus regarding what data to collect in the natural history studies will help to ensure consistency in the evidence reviews. The care provision system must also be inclusive and provide resources to connect the stakeholders, including researchers, clinicians, families, therapeutic developers, treatment providers, and the broader community.

Changes are necessary to close these systemic gaps. Eligibility assessment principles for programs like EIP should be revisited now that prenatal and neonatal diagnoses of an increasing number of genetic conditions associated with NDDs are becoming more common. Presymptomatic diagnoses of highly penetrant NDDs should be considered for automatic eligibility. The assessment tools should be further developed to enable the sensitive evaluation of infant behavior and electrophysiological evidence of brain dysfunction. The treatment paradigm in response to symptoms must be changed to adapt to the new opportunity of proactive intervention and developing therapeutic strategies for a young developing brain. Additionally, in true partnership with families, funding resources should be made available to family support programs irrespective of EIP eligibility, such as online coaching and education on early intervention principles and strategies. Family support groups can also be a resource and provide peer support. Resources should be directed to support families as the systemic gap is being addressed, and more research funding should be allocated to rigorously assess the impact of any interventions applied and consider the age of treatment initiation in the outcomes.

## Data Availability

Data sharing is not applicable.

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
