# Peer review of "An Opportunity to Fill a Gap for Newborn Screening of Neurodevelopmental Disorders"

_2409-515X, 2024, doi:10.3390/ijns10020033_

Round 1

Reviewer 1 Report

Comments and Suggestions for Authors

This commentary highlights an important and under-appreciated gap in services for infants identified with NDDs via newborn sequencing. The experience of the authors via the GUARDIAN study positions them well to speak to these challenges. 

A few additions could strengthen the paper:

1. The authors use the term Early Intervention Programs (EIP) which does not specify the specific service system(s) they are referring to.  The authors appear to be describing the IDEA Part C early intervention system. If so, this should be specified and referenced throughout.

2. Overall the paper needs more references. It is difficult in some places to distinguish between which claims are based on published evidence and which are anecdotal, based on author experiences. Even as an opinion piece, it would be significantly strengthened by disambiguating which elements of the argument are based on established evidence, which reflect the authors' opinions/experiences, and where additional evidence is needed. 

3.  The second paragraph of the Discussion refers to "a critical need for the field to have systems in place to allow families to access genomic screening for genetic NDDs for a period of time before the healthcare infrastructure catches up".  While conceptually true, this sentence (and section) leaves unclear whose responsibility it should be to put "systems in place" to ensure families are supported while treatments are being developed. Is it "the field" (clinical neuroscience/genetics researchers and their funders?) Or EIP programs, with funding from state/federal government? Additional elaboration of the authors' perspective on structural solutions to these identified challenges would greatly strengthen the paper. 

Detailed comments:

1. The Abstract refers to "systemic gaps in medical services". Suggest rewording as the type of early intervention services the authors describe are not typically considered medical services. 

Reviewer 2 Report

Comments and Suggestions for Authors

This commentary argues about the central importance of early intervention programs (EIPs) for children with neuro-developmental disorders identified through newborn screening.

The opening paragraph of the introduction focuses on pilot studies.  The term “pilot studies” can be confusing since it refers to a wide range of activities (e.g., prospective tests of screening in public health programs, consent-based activities that may happen outside of newborn screening programs).  It might help to clarify the specific context for pilot studies in this paragraph.

Line 43 underplays how little is often known about the benefits and harms of early detection.  Saying “…this group of conditions have generally not been considered for NBS due to the lack of completely effective treatment.” minimizes that there may be no known effective treatment and that some families might not want to know.  In addition, for some conditions, predicting the degree of involvement or phenotype is not possible.  This lack of certainty should be woven into the commentary so that it does not imply that we know that early intervention makes a difference but that it is a reasonable thing to do.

The discussion of how Early Intervention Programs empowers families should have additional context and nuance.  In the US, states choose how to implement their programs with funding from the federal government.  There should be a discussion of the IDEA and how these services may not be uniform or even accessible to all families, with variation in availability even with states.  Furthermore, as the number of children become eligible, unless there is a change in funding, the available resources will be further stretched or not available.  It is simply not sufficient to say that more children should be eligible and expect that this alone will lead to better access and hopefully better outcomes.  The conclusion makes a clear argument for the types of programs that should be available and mentions challenges in access.  The commentary would be much stronger with specific recommendations for how EIPs should be funded and operate.

Round 2

Reviewer 1 Report

Comments and Suggestions for Authors

The authors have added recommendations to the discussion section that significantly strengthen the paper. The additional references are also helpful. With these revisions, I am confident that this paper will make a strong contribution to the literature. 

Reviewer 2 Report

Comments and Suggestions for Authors

Thank you for addressing the issues raised in my previous review.  The discussion of the IDEA will be helpful to policy makers who read this commentary.